# Spontaneous EBV-Reactivation during B Cell Differentiation as a Model for Polymorphic EBV-Driven Lymphoproliferation

**DOI:** 10.3390/cancers15123083

**Published:** 2023-06-07

**Authors:** Matthew A. Care, Sophie Stephenson, Roger Owen, Gina M. Doody, Reuben M. Tooze

**Affiliations:** 1Division of Haematology and Immunology, Leeds Institute of Medical Research, University of Leeds, Leeds LS9 7TF, UK; m.a.care@leeds.ac.uk (M.A.C.); s.j.stephenson@leeds.ac.uk (S.S.);; 2Bioinformatics Group, School of Molecular and Cellular Biology, University of Leeds, Leeds LS2 9JT, UK; 3Haematological Malignancy Diagnostic Service, Leeds Teaching Hospitals NHS Trust, Leeds LS9 7TF, UK; r.owen@nhs.net

**Keywords:** Epstein-Barr virus, B-cell, germinal centre, plasma cell, latency, reactivation

## Abstract

**Simple Summary:**

Epstein-Barr virus (EBV) infects a high proportion of the human population. After initial infection, EBV is maintained in healthy individuals in an inactive latent state in memory B-cells of the immune system. Reactivation of EBV from latency can drive B-cell tumours. EBV mimics immune signals to drive B-cells into specific activation states, expanding the virally infected B-cell population. Here we use a laboratory model of the human memory B-cell immune response to study the frequency and patterns of B-cell activation that occur upon spontaneous EBV reactivation. To do this we use single cell analysis of RNA expression. Consistent with the high prevalence of latent EBV infection we find evidence of EBV in cells from all eight healthy adult donors tested. While the extent varies between donors, we identify four recurrent cell states that EBV-associated B-cells adopt upon viral reactivation. Our results establish a model for studying polymorphic EBV-driven B-cell lymphoproliferation.

**Abstract:**

Epstein-Barr virus (EBV)-driven B cell neoplasms arise from the reactivation of latently infected B cells. In a subset of patients, EBV was seen to drive a polymorphous lymphoproliferative disorder (LPD) in which B cell differentiation was retained. In this work, spontaneous EBV reactivation following B cell mitogen stimulation was shown to provide a potential model of polymorphic EBV-driven LPD. Here, we developed an in vitro model of plasma cell (PC) differentiation from peripheral blood memory B cells. To assess the frequency and phenotypes of EBV-associated populations derived during differentiation, we analysed eight differentiations during the PC stage with a targeted single-cell gene expression panel. We identified subpopulations of EBV-gene expressing cells with PC and/or B cell expression features in differentiations from all tested donors. EBV-associated cells varied in frequency, ranging from 3–28% of cells. Most EBV-associated cells expressed PC genes such as *XBP1* or *MZB1,* and in all samples these included a quiescent PC fraction that lacked cell a cycle gene expression. With increasing EBV-associated cells, populations with B cell features became prominent, co-expressing a germinal centre (GC) and activating B cell gene patterns. The presence of highly proliferative EBV-associated cells was linked to retained *MS4A1/CD20* expression and *IGHM* and *IGHD* co-expression, while *IGHM* class-switched cells were enriched in quiescent PC fractions. Thus, patterns of gene expression in primary EBV reactivation were shown to include features related to GC B cells, which was also observed in EBV-transformed lymphoblastoid cell lines. This suggests a particular association between spontaneously developing EBV-expansions and IgM+ IgD+ non-switched B cells.

## 1. Introduction

The majority of the adult human population is predicted to be latently infected with the Epstein–Barr virus (EBV) [1]. EBV establishes latency in its host primarily in memory B cells, and a balanced immune control sustains this state [2]. The balance between viral latency and immune control allows for a sustained viral persistence. In the face of reduced T-cell mediated surveillance, reactivation from latency is associated with EBV-driven lymphoproliferative disorders (LPDs). These may be polyclonal, with a retained plasmacytic differentiation, or they may recapitulate features of aggressive lymphoma, including subsets of classical Hodgkin lymphoma, diffuse large B cell lymphoma, and Burkitt lymphoma, both in immune-competent and immune-suppressed or immune-deficient states [1,3]. The association of EBV with malignancy is not restricted to B cell lineage and is also implicated in the pathogenesis of some NK/T-cell lymphoma, undifferentiated nasopharyngeal carcinoma, and rare gastric cancers [1].

EBV is primarily maintained in a latent state in memory B cells [4]. Several distinct memory B cell subsets have been defined in humans, which broadly can be divided into the class-switched memory of IgG or IgA isotypes, IgM+ IgD-, and IgM+ IgD+ non-switched memory (NSM) B cells [5,6]. In the chronic carrier state, EBV viral loads are the highest in class-switched memory B cells [7]. While it has been suggested that EBV is excluded from NSM B cells [8], quantitative analyses that have tested individuals have documented significant EBV viral loads in highly pure fractions of CD27+ IgM+ IgD+ NSM B cells [7]. In contrast, EBV is present at only very low levels in naïve B cells. Indeed, in some individuals IgM+ IgD+ NSM B cells can be the primary reservoir for EBV latency and this is particularly the case in immune deficient states where conventional memory B cells are not established [9]. In vitro, EBV can infect and immortalize naïve B cells, NSM, and switched memory B cells with equivalent efficiency [10]. Hence, the preferential retention of EBV in memory B cell subsets is considered to relate to how the virus interacts with the B cell compartment during in vivo immune responses [1,11].

Stimuli for reactivation from latency include immunological signals driving B cell activation, such as a B cell receptor ligation [12]. Differentiation to the plasma cell state (PC) has been linked to entry in the viral lytic replication [13]. As the interaction between virus and host progresses from primary infection to latency (and potentially to reactivation), distinct programmes of gene expression are executed from the viral episome [1,14]. Encoded EBV proteins regulate the expression of alternate viral programmes, viral replication, and entry into the lytic cycle, as well as controlling the activation and differentiation state of the host B cells. For example, the proliferative expansion of latently infected B cells is driven by latent membrane protein 1 and 2 (LMP1 and 2) expression. These simulate B cell activation signals associated with a T-dependent immune response in which LMP2 mimics signals derived from a B cell antigen receptor [15]; LMP1 mimics signals from the primary TNF superfamily receptor protein, CD40, which is associated with T cell help for B cell activation [16]. LMP1 and LMP2 can synergise to transform B cells independent of other EBV components [17]. EBV-mediated transformation of mature B cells to generate lymphoblastoid B cell lines (LCLs) has provided model systems for LMP1- and LMP2-associated latency states. These are the exhibited features of B cell activation and partial PC differentiation, reflecting both signalling from LMP1/LMP2 and a clamp that EBV nuclear proteins place on the epigenetic state of the activated B cell [18,19,20]. Importantly, LCLs differ from cell lines derived from patients with EBV-driven lymphoma, such as Burkitt lymphoma, wherein other pathogenic mutational events contribute to the specific neoplastic state [21,22].

Recently, single-cell gene expression analyses of LCLs have begun to shed further light on the transcriptional programmes of EBV-transformed cells [23,24]. These demonstrated that LCLs are comprised of distinct but interconnected subpopulations of cells. Some cells show features suggestive of an abortive germinal centre (GC), such as the expression programme, in which signatures related to LMP1/CD40-like signals mimic the physiological counterpart of a light zone (LZ) B cell, while other B cells show high expression of proliferative gene expression programmes, as observed in the GC dark zone (DZ) B cells. Additionally, subsets of cells in LCLs have been shown to exhibit features of PC differentiation. Notably, similar patterns of GC-like gene expression were observed in time-resolved gene expression analysis of bulk B cell populations following acute infection with laboratory EBV strains [25,26]. 

EBV reactivation from latency following exogenous mitogenic signals is a well-established phenomenon [12,27]. However, to our knowledge there is relatively little known regarding patterns of gene expression at the single-cell level in spontaneously emerging EBV reactivation in primary human B cell cultures. In the past, researchers, including us, have developed models for in vitro B cell differentiation in order to study the generation and control of human PC differentiation [28,29,30]. These models were combined with detailed expression analysis to provide an opportunity to investigate EBV reactivation in differentiated memory B cells of healthy donors. Here, we describe the patterns of gene expression linked to spontaneous EBV reactivation in differentiated B cell populations, identifying features that link to the differentiation state, and are consistent with recently described features in EBV LCLs.

## 2. Methods

### 2.1. Reagents

The following reagents were used: IL2, IL21, and IL6 (Miltenyi, North Rhine-Westphalia, Germany); multimeric APRIL (R&D, Santa Clara, CA, USA); F(ab’)_2_ fragments of goat anti-human IgA, IgM, and IgG (Jackson ImmunoResearch, West Grove, PA, USA); chemically-defined lipid mixture 1 (200×); and MEM amino acids (50×) from Sigma (St. Louis, MO, USA). 

### 2.2. Donors and Cell Isolation

After informed consent, peripheral blood was provided by 8 healthy individuals (age range 25–55 years, without evidence of intercurrent infection or known medical condition). Lymphoprep (Abbott, Maidenhead, UK) and density gradient centrifugation was used to isolate mononuclear cells. A two-step negative selection protocol was applied to isolate memory B cells using the memory B cell isolation kit (Miltenyi). Here, total B cells were isolated by negative selection followed by naïve B cell depletion using anti-CD23-biotin antibody (Miltenyi 130094510) and anti-biotin microbeads to leave untouched memory B cells. 

### 2.3. Cell Cultures

B cell differentiation cultures were performed in Iscove’s modified Dulbecco medium (IMDM) with supplements of Glutamax and 10% heat-inactivated foetal bovine serum (HIFBS, Invitrogen, Waltham, MA, USA). On day 3 of culturing, lipid mixture 1 and MEM amino acids were added at 1× final concentration.

From days 0 to 3, cultures were maintained in 24 well plates at 2.5 × 10^5^/mL. B cells were activated on γ-irradiated CD40L expressing L-cells (6.25 × 10^4^/well) supplemented with IL2 (20 U/mL), IL21 (50 ng/mL), and F(ab’)_2_ goat anti-human IgA/M/G (2 µg/mL).

From days 3 to day 6, cells were removed from the CD40L L-cell layer (day 3) and then cultured at a starting density of 1 × 10^5^/mL with IL2 (20 U/mL) and IL21 (50 ng/mL).

From days 6 to day 13, cells were transferred (day 6) into media supplemented with IL6 (10 ng/mL), IL21 (10 ng/mL), and multimeric APRIL (100 ng/mL) at a starting density of at 1 × 10^6^/mL.

On day 13, cells were reseeded in half the volume of fresh media used on day 6 and supplemented with IL6 (10 ng/mL) and multimeric APRIL (100 ng/mL). 

On days 20–24, dead cells were removed by lymphoprep density gradient centrifugation. Live cells were placed in fresh media for 72 h. Cells were then cryopreserved.

### 2.4. Flow Cytometric Analysis

Flow cytometry was performed with 6-color direct immunofluorescence detection using a Cytoflex LX (Beckman Coulter, Brea, CA, USA). Staining was performed with the following antibodies: CD19 PE (LT19, Miltenyi 130113169), CD138 APC (44F9, Miltenyi 130117395), CD20 e450 (2H7, Thermo Fischer 48020942) (Thermo Fischer, Waltham, MA, USA), CD27 FITC (M-T271, BD 55540), and CD38 PECy7 (HB7, BD 335825). Isotype-matched antibodies were used as controls (IgG1—PE Miltenyi 130113200; IgG1—APC Miltenyi 130113196; IgG2b—eFlour450 Thermo Fisher 48473282; IgG1—FITC BD 348808; IgG1—PECy7 Miltenyi 130113196,). The zombie UV fixable viability kit (Biolegend 423108, San Diego, CA, USA) was used to exclude dead cells. FlowJo version 10 (BD Biosciences, Franklin Lakes, NJ, USA) and Prism 9 (GraphPad, San Diego, CA, USA) with Spearman correlation statistical analysis were applied for analysis.

### 2.5. Single-Cell RNAseq 

Single cell experiments were carried out using the BD Rhapsody express single-cell analysis system. 

Cryopreserved cells were thawed, and dead cells removed via lymphoprep density gradient centrifugation. Cells were enumerated on a haemocytometer.

The cells from individual donors were labelled with a unique sample tag identifier using the BD single-cell multiplexing kit. Donors were pooled in equal ratios, with the aim of loading 4000–5000 cells per donor onto a cartridge. Donors 1–4 and 5–8 were run on independent cartridges using a custom designed 701 gene panel (Appendix A).

Manufacturers protocols were followed throughout.

Libraries were sequenced on a Novoseq6000, PE 150, S4 flow cell with a target of 152 Gb of raw data per experiment. Data sets are available via the Gene Expression Omnibus with accession number GSE232418.

### 2.6. Data Processing 

FASTQ files were processed via the BD Rhapsody targeted analysis pipeline (v1.10.1) on Seven Bridges (https://igor.sbgenomics.com/ (accessed on 17 June 2022)) using the recommended settings. The resultant DBEC_molsPerCell and Sample_Tag_Calls files were imported into R (v4.1.2) and processed with the Seurat package (v4.2.0) [31,32]. Data from runs 1 and 2 were analysed independently. In each case, filtered cells were annotated as undetermined or multiplet. Genes were filtered if expressed <0.1% of cells, and cells were filtered if expressing a number of genes <1st quartile. The remaining data was log normalised. 

### 2.7. Cell Subsets 

The normalised expression data was subdivided based on gene expression. Any cell expressing an EBV gene (i.e., *BALF1, BHRF1, BLLF1.gp350, BZLF1, EBER1.2, EBNA1, EBNA2, EBNA3A, EBNA3B.C, LMP1, LMP2A.2B*) > 1 standard deviation (SD) was assigned to the EBV subgroup.

### 2.8. Clustering and Differential Gene Expression Analysis 

The filtered normalised data was assessed using the jackstraw approach (prop.freq = 0.05, num.replicate = 100). The highest significant principal component number (maxPC) was used to construct a KNN graph for dimension reduction (FindNeighbors: dims=maxPC), which was then clustered using the Louvain algorithm with multilevel refinement (FindClusters: resolution = 0.5, n.start = 25, algorithm = 2) and visualised as a UMAP (RunUMAP/DimPlot: dims = maxPC). Marker genes for each cluster were identified by comparing gene expression within a cluster versus all other cells using FindAllMarkers (test.use = “MAST”, only.pos = TRUE, min.pct = 0.01, logfc.threshold = 0.25) with the MAST package (v1.20.0). 

### 2.9. Gene Signature Data and Enrichment Analysis 

A dataset of 20,707 gene signatures was created by merging signatures downloaded from http://lymphochip.nih.gov/signaturedb/ (SignatureDB, accessed on 13 May 2020), http://www.broadinstitute.org/gsea/msigdb/index.jsp MSigDB V7.4 (MSigDB C1–C8 and H; excluding C5, accessed on 9 July 2021), Human CORUM complexes with > 2 genes (http://mips.helmholtz-muenchen.de/corum/#download (accessed on 13 May 2020)), UniProt keywords (parsed XML; http://www.uniprot.org/downloads (accessed on 9 July 2021)), and 18 papers (PMIDs: 12975453, 15550490, 19412164, 20725040, 21179087, 23563690, 23584089, 23584090, 23700391, 23871637, 24138885, 24220563, 24336226, 24644022, 25800755, 25822800, 28735890, and 32603407). A gene ontology gene-set was created using an in-house python script. This parses a gene association file (http://geneontology.org/page/download-annotations (accessed on 9 July 2021)) to link genes with ontology terms, then uses the ontology structure (.obo file; http://geneontology.org/docs/download-ontology/ (accessed on 9 July 2021)) to propagate these terms up to the root. The resultant gene-set contained 22,865 terms. The gene-ontology and gene-signature sets were merged to give a final signature set of 43,572 terms. Gene signatures that lacked any overlap with the targeted 701 gene panel were excluded. Enrichment of clusters for signatures was assessed using a hypergeometric test, with the draw being the cluster-marker genes. The successes were the signature genes, and the population was the target panel genes after filtering. 

### 2.10. Significance of Cell Groups 

Cells were split into groups based on their expression patterns. Cells were stratified based on their *MS4A1 (CD20)* and *IGHD/IGHM* (constant region of heavy chain of *IgD* or *IgM*) expression. For the comparison of cell cycle and EBV gene expression, cells expressing any of *CDK1* (cell division cycle 2, G1 to S and G2 to M)/*FOXM1* (Forkhead Box M1/M-phase phosphoprotein 2)/*MKI67* (marker of proliferation Ki-67) were assigned to the CellCycle+ve group. The CellCycle+ve/−ve groups were then stratified based on their expression of *LMP1* and by the number of EBNA genes (*EBNA1, EBNA2, EBNA3B.C*) ranging from 0 (EBNA−ve) to all 3 genes being expressed (EBNA#3). In all cases, cells with normalised. Counts > 0 were deemed to show expression of the corresponding gene. For each group, the *p*-value of enrichment/depletion was calculated using a Fisher’s exact test.

### 2.11. Data Visualisation 

Figures were generated using the output of Seurat, with ggplot2 (v3.3.6), as well as python (v3.9.15) and the package seaborn (v0.11.2; matplotlib v3.5.1).

### 2.12. Ethical Approval

Approval for this study was provided by the UK National Research Ethics Service via the Leeds East Research Ethics committee, approval reference 07/Q1206/47 and IRAS project ID 187050.

## 3. Results

### 3.1. A Targeted Single-Cell Expression Panel for B Cell and PC Analysis

To assess the potential contribution of EBV-associated populations and their expression states during in vitro B cell differentiation, we developed a custom targeted expression panel for 701 genes on the BD Rhapsody platform (Appendix A). To select the panel genes, we took advantage of our previous work, which generated gene correlation networks from bulk RNAseq data across in vitro B cell to PC differentiation and gene correlation patterns in diffuse large B cell lymphoma. The aspects of panel selection are summarised in Appendix A. Briefly, they included 260 genes selected based on consistent network information content, 140 genes manually selected across network modules to capture additional informative hub genes, and 301 genes selected from prior literature across B cell, cell cycle, and neoplastic plasma cell biology. Moreover, marker genes for T- and myeloid lineages were included. Lastly, the following 11 gene targets were selected from the EBV genome: *EBER1/2*, *EBNA1*, *EBNA2*, *BHRF1*, *EBNA3A*, *EBNA3B/C*, *LMP1*, *LMP2A/2B*, *BLLF1*, *BZLF1*, and *BALF1*. 

### 3.2. Assessing Overall EBV Contribution in Late B Cell Differentiation

Given the predicted high frequency of latent EBV infection in the otherwise healthy adult population [33], we reasoned that the differentiation of memory B cells from healthy donors would provide a means to assess the frequency and pattern of EBV reactivation upon B cell differentiation. We defined healthy adult donors only on the basis of lack of known intercurrent infection or a declared medical condition. We did not assess EBV status prior to differentiation. To explore the potential contribution of EBV-associated populations amongst in vitro differentiated cells, we purified memory B cells from a total of eight donors (DN) divided between two experiments with four independent donors each. We differentiated these cells to beyond day 20 in the context of APRIL-supported survival conditions. APRIL was used to support PC differentiation because this mimics an important element of the in vivo niche conditions that maintain long-lived PCs [28]. Differentiated cells were evaluated with single-cell gene expression analysis between days 23–27 of differentiation, a time point where a differentiated PC state was established for the population average (Figure 1 and Appendix A) [28,29,30]. We selected this time point as cells with either a retained B cell state or high levels of cell cycle gene expression would be readily differentiated as outliers from the population average, predicted by our prior bulk population gene expression studies [28,29,30]. Flow cytometric analysis of the differentiated populations at the test time point for all eight donors showed broadly similar phenotypic patterns and failed to show clear evidence of an aberrant lymphoblastoid phenotype (Appendix A). However, the culture media for the cells from donor 4 (DN4) exhibited rapid media acidification, making it suspicious of potential transformation. 

We performed two independent analyses of single-cell gene expression on these sets of samples (Figure 2) and focused on the patterns of expression of EBV-associated B cell populations. Samples were multiplexed and a total of 30,851 cells were recovered across both experiments reflecting 17,188 and 13,663 cells from the two experiments. Expression of 625 genes from the 701 target gene panel was detected. The most abundant transcripts were *IGKC*, *IGLC1,* and *JCHAIN* consistent with the PC differentiation state. Other abundantly expressed genes linked to the PC state included *TXNDC5, MZB1, XBP1, ATF4, IRF4, SLAMF7* (*CD319*), and *TNFRSF17 (BCMA)*. To maintain the experimental replication throughout, the two data sets were independently analysed, combining cells from all four donors in each experiment or assessing cells from each donor individually. 

Considering the expression of any single EBV gene to be evidence of EBV association, the contribution of EBV-associated B cells ranged from 3% to 28% of cells across the samples (mean = 0.0975, stdev = 0.0785, median = 0.075). DN4 suspected of EBV transformation based on growth characteristics was confirmed as having the highest proportion of EBV-associated B cells (Figure 2b). Analysis of the two independent experiments gave similar results (Figure 2c). Both sets of samples resolved into eight clusters (Figure 2d). In both instances, two primary groups of cells were resolved as the larger encompassing clusters C0-2+4 (experiment 1) or C0-4+7 (experiment 2). Markers of the PC state, including gene such as *MZB1*, *IRF4,* and *XBP1*, were expressed across all clusters, while the separation of cell clusters was linked to differences in the expression of marker genes of the B cell state (*MS4A1, LMO2*) and cell cycle (*CDK1, FOXM1*). EBV genes were primarily localised in clusters associated with B cell and cell-cycle features (Figure 3). Thus, while a PC state characterizes the majority of cells, this is accompanied by a variable proportion of cells retaining B cell and cell-cycle features. The latter features are associated with cell clusters also showing evidence of EBV-associated gene expression.

### 3.3. EBV-Associated Cells Separate Based on Differentiation and Cell-Cycle State

To further assess the EBV-associated cells, we repeated the analysis for the two independent experiments, focusing on all cells identified by expression of EBV genes. Experiment 1 resolved into 5 cell clusters (Figure 4a) and experiment 2 into 6 (Figure 4b). The contribution across the four donors varied in each experiment but followed a general pattern such that donors with larger absolute numbers of EBV-associated cells contributed to a wider range of cell clusters, while those with small numbers of EBV-associated cells showed a restricted contribution to one or two clusters.

For EBV transcripts, the broadest detection across multiple cell clusters was observed for *EBNA1, EBNA2, EBNA3B.C*, and *BHRF1*. More restricted expression was observed for *LMP1, LMP2A,* and *BLLF1.gp350* (Figure 5). *EBER1/2* expression was not adequately detected; therefore, conclusions regarding cells in latency 0 cannot be drawn. While PC marker genes such as *MZB1, XBP1, IRF4,* and *PRDM1* were broadly expressed, restricted expression was observed for features of the B cell state, such as surface markers (e.g., *MS4A1/CD20, CD40, CD48* and *CD83)*, features of cell proliferation (e.g., *CDK1* and *FOXM1)*, transcription factors (e.g., *BATF, IRF8, LMO2, MYC,* and *SPIB),* and other regulators, such as *MIR155HG*.

To further assess the relationship of resolved cell clusters (designated e.g., E1_EBV_C0) to known patterns of gene expression, we applied gene signature and ontology analysis to the cluster marker genes in each experiment (Appendix A). The signature enrichments amongst cluster marker genes for both experiments support similar separation of cell clusters. These are characterised by predominantly plasmablast/PC features (E1_EBV_C0&C3 and E2_EBV_C2&C4) and those related to cells with retained B cell features (E1_EBV_C1&C2 and E2_EBV_C0&C1&C5). The former quiescent cell clusters characterised by PC related expression features (E1_EBV_C0 and E2_EBV_C2) were separated from proliferative fractions with predominant plasmablastic features (E1_EBV_C3 and E2_EBV_C4). The clusters with B cell features were separated based on patterns of expression related to CD40/NFkB signalling and the GC LZ (E1_EBV_C2 and E2_EBV_ C5) as opposed to activated B cells with dominant cell-cycle features and a relationship to GC DZ B cells in both experiments (E1_EBV_C1 and E2_EBV_C1) [34,35]. A separate subset of cells enriched for IFN response signatures and features related to pre-memory B cells were identified in experiment 2 (E2_EBV_C0). Thus, across both experiments, recurrent cell states were identified that share features relating to GC biology and the plasmablast to PC transition.

### 3.4. Cell Cycle and MS4A1/CD20 Expression Are Linked to EBV-Gene and IgM/IgD Expression

We noted that the pattern of EBV gene expression was skewed in relation to the cell-cycle state. For example, the proliferative and GC DZ-like cluster in experiment 1, E1_EBV_C1, included the EBV marker gene combination of *EBNA1*, *EBNA2,* and *EBNA3B/C,* but not *LMP1*. This pattern is in keeping with the latency IIb state [14]. Therefore, to further assess the relationship between the EBV gene expression and cell-cycle state, we considered an alternate approach. We defined a cell as being in cell cycle through the expression of any one of three marker genes *(CDK1, FOXM1,* and *MKI67)*, and then examined the relative likelihood of EBV gene expression in such cells (Figure 6a). We focused on the presence or absence of *LMP1* expression and on the number of *EBNA* genes expressed. This analysis provided a simplified assessment of the potential subdivision between latency III or latency IIa, in which *LMP1* is expressed and latency-IIb in which multiple *EBNAs* are expressed without *LMP1* [14]. Across both experiments, we found a highly significant association between the expression of multiple *EBNA* genes and the cell cycle-related gene expression. This was significantly more pronounced in the absence of *LMP1* expression. Thus, expression of multiple *EBNA* genes without *LMP1* showed the strongest association with cell cycle, which is broadly in keeping with a latency IIb-like cell population.

The targeted panel included the detection of heavy chain isotypes for IgM, IgD, and IgA. While IgG targeting could not be designed in the panel, a lack of other isotypes was assumed to primarily reflect IgG switched cells. We noted that the expression of *IGHM* and *IGHD* was heavily skewed toward clusters in which cells retained *MS4A1/CD20* expression (E1_EBV_C2 and E2_EBV_C1). To assess the significance of this association, we tested the likelihood of observing *MS4A1/CD20* expression in relation to the presence of *IGHM* and/or *IGHD,* or neither of these heavy chains in individual cells (Figure 6b). This demonstrated a highly significant association across both experiments. The reciprocal was also the case, indicating that cells lacking *MS4A1/CD20* expression were significantly skewed to either an *IGHM* single-positive or a class-switched state, as inferred by the lack of both *IGHM* and *IGHD* expression. Since the pattern of *MS4A1/CD20* expression overlapped with that of cell cycle genes (Figure 5), we considered whether the expression of *MS4A1/CD20* was also significantly associated with the cell-cycle state, as defined by the expression of three marker genes (i.e., *CDK1, FOXM1, MKI67*). This also showed a highly significant correlation between *MS4A1/CD20* expression and the positive cell-cycle state (Figure 6c). Therefore, we concluded that the presence of proliferating cells in the culture was linked to specific patterns of EBV gene expression, as well as to the presence of cells expressing *MS4A1/CD20*. Moreover, the expression of *MS4A1/CD20* was found to be associated with the retained expression of *IGHM* and *IGHD.*

Since CD19 and CD20 expression was included in the original phenotypic analysis of the differentiating B cells, we interrogated the flow cytometric phenotypes back gating onto CD19^hi^ CD20^hi^ populations in the differentiations (Appendix A). These showed varying fractions between 2.5% and 19.6% of cells. We then assessed the correlation between the fraction of CD19^hi^ CD20^hi^ expressing cells, and the number of EBV-associated cells identified using single cell expression analysis showed a significant correlation R = 0.7789, p = 0.0295 (Spearman rank).

Thus, the overall pattern of cellular gene expression in EBV+ cells was found to be significantly associated with the nature of the underlying B cell state. The emergence of cell populations with features of the retained B cell state, as exemplified by *MS4A1/CD20* expression, was strongly linked with an IgM+ IgD+ non-class-switched state and was biased from class-switched cells. 

### 3.5. Donor-Specific Patterns of EBV-Associated B Cells

Each donor contributed distinct fractions of the EBV-associated cell pool and the combined analysis may mask donor specific differences. Therefore, we assessed cell clustering and gene expression for each donor’s EBV-associated cells independently. Overall, between two-and-five cell clusters were resolved for each donor (Appendix A). To provide a systematic assessment, we considered signature enrichment for marker genes that were associated with the cell clusters for each donor (Appendix A). As in the combined experimental analyses, cell clusters were separated based on the extent of the expression of PC and B cell features, as well as the activation state and cell-cycle genes across individual donors. A recurrent feature across all donors was the presence of a PC-like population, with a low (or lack) of expression of B cells, cell cycle, and activation markers. This was the predominant pattern for the two clusters resolved, e.g., in DN5 with the lowest EBV-associated cell number. As the absolute EBV-associated cell number increased (DN5 < DN7 < DN2 < DN1 < DN3 < DN6 < DN8 < DN4), clusters with B cell-related gene expression were resolved. These separated further based on patterns of expression related to B cell activation, cell growth, and the GC LZ state, as well as those dominated by the expression of genes related to the cell cycle and hence the GC DZ (Appendix A). 

DN4 and DN8 provide the index examples in each experiment for differentiations with the highest EBV-associated cell number. DN4 EBV-associated cells resolved into four clusters (Figure 7). DN4_EBV_C0 combined features of activated and GC DZ B cells, along with MYC and E2F target genes and cell cycle features. DN4_EBV_C1 combined features of the cell cycle and the plasmablast state. DN4_EBV_C2 was associated with features of the core B cell state and NFkB signalling; it was also enriched for genes related to GC LZ B cells, but was depleted of cell cycle features. DN4_EBV_C3 was linked to features of the PC state and lacked features of cell activation or cell cycle. 

DN8 repeated this general pattern but subdivided into five clusters. These separated DN8_EBV_C0 with PC signatures but not the cell cycle from DN8_EBV_C1 with plasmablastic features and the cell cycle, as well as MYC- and E2F-associated signature enrichment. DN8_EBV_C2 was associated with B cell-related gene expression, IRF4 target genes, and features of pre-memory B cells. DN8_EBV_C3 was linked to features associated with activated B cells, MYC and E2F targets, and cell-cycle signatures; it was also enriched for genes related to GC DZ B cells. Finally, DN8_EBV_C4 was linked to features of GC LZ B cells, activated B cells, NFkB signalling, and MYC target genes. 

Next, we assessed the relationship of EBV gene expression to the donor-level cell clusters (Appendix A). This was most informative for DN4 and DN8, as it demonstrated that resolved clusters separated cells with a latency IIb-like state (i.e., stronger *EBNA1, EBNA2,* and *EBNA3B.C* but weaker *LMP1*) into clusters with GC DZ-like features (DN4_EBV_C0 and DN8_EBV_C3) from cells with an overall pattern related to latency III or IIa (stronger *LMP1* with EBNAs) into clusters with GC LZ-like features (DN4_EBV_C2 and DN8_DBV_C4). Moreover, clusters linked to quiescent PC-like states (DN4_EBV_C3 and DN8_EBV_C0) showed the weakest detected EBV gene expression, which for DN4 was most strongly detected for the anti-apoptotic EBV gene *BHRF1*.

Hence, the systematic analysis of the EBV-associated B cells across eight independent donors resolved repeated patterns that separated distinct differentiation and activation states. At low EBV-associated cell numbers, the recurrent pattern of the PC state showed little evidence of a cell cycle. A higher EBV-associated cell number were linked across different donors to a wider spectrum of cell states, resolving patterns of gene expression that could be related to B cell states in the GC and plasmablast to PC differentiation, and associated with distinct patterns of EBV gene expression.

## 4. Discussion

EBV-driven B cell proliferations represent a major clinical burden. Increasingly, classifications of B cell neoplasms have appreciated the wide spectrum of neoplastic and pre-neoplastic patterns. Setting aside the frankly neoplastic spectrum, this includes the expanding range of EBV-driven lymphoproliferations that occur in documented immunodeficiency or in the context of immune senescence, wherein the presence of EBV-associated B cells is linked to a range of B cell morphologies and differentiation states. Such polymorphous proliferations may further vary, exhibiting either tissue destructive patterns or relatively retained tissue architecture, and polyclonality through to oligoclonal or clonal dominance [3]. An implicit concept in these entities is that the immune surveillance of the EBV-latent state has been undermined and, thus, the reactivation of EBV has been presumptively driven by an immune stimulus [2]. 

LCLs have served as valuable tools to understand the biology of EBV-driven quasi-neoplastic cell states. Here, we described a complementary set of findings that relate to the spontaneous emergence of EBV-driven B cell populations amongst in vitro differentiating human memory B cells from healthy donors (Appendix A). Consistent with the high prevalence of EBV in the adult population, we found that all eight donors generated some component of EBV-associated B cell/PC populations upon differentiation of memory B cells in the context of an APRIL-driven survival signal for the PC state. These populations remain a minor fraction of the total differentiated PC pool in most donors but can establish dominance in some under these differentiation conditions. The spontaneous emergence of EBV-associated B cells exhibited related but varied patterns between individual donors when observed as a snapshot after PC differentiation with scRNAseq. In donors with the smallest EBV contribution, viral transcripts were detected in quiescent PC-like cell fractions and were primarily class-switched. Expanded EBV-driven populations were associated with the emergence of cells with mixed PC- and B cell-like features, related to those in LCLs [23], and an association with IgM and IgD co-expression. Such cells occupy a range of activation and cell-cycle states, which recapitulate aspects of gene expression that map onto GC B cell states.

In donors with higher EBV-associated cell numbers, the proportion of cells in the cell cycle increased. This was associated with two expression states, one of which was plasmablastic and the other of which retained B cell features. We reason that based on the similarity to features observed in LCLs [23,24], EBV is driving the retained B cell features in the latter population and restraining differentiation to the plasmablast/PC state. We speculate that this may be driven by a combination of sustained signalling and the ability of EBV proteins to control the epigenome of the B cell [18,19,20]. Such possibilities will be interesting to explore in future work. 

A limitation of our current study is the snapshot nature of analysis at a single time point. This time point was selected because of the contrast provided by the bulk population of differentiated and quiescent PCs as opposed to any proliferative or B cell phenotype retention that might be observed. We acknowledge the inherent limitation of a single time point, meaning that we cannot reach conclusions regarding either preceding or subsequent patterns of expression associated with EBV-associated cells in these experiments. Nonetheless, the heterogeneity of cell states observed between donors proved informative for the range of EBV-associated cell states that were observed in the model, providing an impetus for further exploration of the nature of EBV gene expression over time.

EBV-associated gene expression included a detectable latency III-like pattern that was present primarily in the context of *MS4A1/CD20* expressing cells, showing evidence of CD40-like signalling and aspects of GC LZ programmes. These sat alongside highly proliferative cells with patterns of expression characterised by detection of *EBNA1* and *EBNA3B/C,* but not *LMP1*. This was consistent with the latency IIb expression pattern [14]. Recently, Luftig et al. [23,24] analysed the expression programmes in LCL using scRNAseq. Further, the patterns of expression were mapped back onto features of the GC reaction and cell-cycle state. These analyses demonstrated heterogenous patterns of EBV gene expression linked to cell state transitions and expression patterns that also mimic features of GC B cell compartments that separate LZ-like gene expression with NFkB pathway activation, from highly proliferative cell fractions related to GC DZ-like expression patterns lacking NFkB-related gene expression [24]. 

Our data supports a model in which there is a transition between related cell states in spontaneously arising EBV-driven B cell populations seen at the single cell-level. On the one hand, a latency III programme in which *LMP1* is associated with expression patterns that overlap with the GC LZ growth programme; on the other hand, a latency IIb-like programme in which *LMP1* expression and NFkB pathway activation is lacking, and *EBNA* gene expression is linked to a hyperproliferative state that mimics GC DZ features. We cannot exclude the possibility that the link between proliferation and a latency IIb-like programme in our data is related to the de novo infection that occurs following earlier lytic reactivation during the differentiation process. However, we favour the suggestion that there is intrinsic transition in EBV latency and associated B cell states related to LZ/DZ cycling, as demonstrated in LCLs [24]. A notable feature of the spontaneously expanded EBV-driven B cell populations we observed here is their highly significant association with the co-expression of *IGHM* and *IGHD,* as well as the reciprocal depletion of this phenotype from the more quiescent PC clusters. This suggests two possibilities. One is that the PC clusters are derived from the LZ/DZ-like proliferating B cell fraction with an accompanying cell cycle exit; the second is that the PC clusters arise from independent reservoirs of latent IgM+ IgD−ve or class-switched memory B cells. In donors generating small fractions of EBV+ cells, IgM+ IgD−ve and class-switched EBV+ cells predominate and show PC features, and the LZ/DZ-B cell-like populations are not distinguished. Therefore, we favour a model of independent origin of these cells in many donors. However, future experiments are needed to directly address the relative origins of these populations. 

Contrasting with the PC-like fractions, the LZ/DZ-like populations that most resemble LCLs are highly enriched for cells expressing *IGHM* and *IGHD*. Since the differentiations derived from memory B cell-enriched fractions, we reason that these either arise from IgM+/IgD− memory B cells that re-express *IGHD* in the context of EBV reactivation or derive selectively from IgM+/IgD+ memory B cells. We acknowledge that a limitation of our study is that we have no direct evidence to differentiate between these possibilities and cannot exclude the possibility of a contribution from contaminating naïve B cells or lytic reactivation and acute infection. However, IgM+ IgD+ NSM B cells have been previously identified as a reservoir of latent EBV infection, which can vary in frequency between healthy carriers [7]. While the relative frequency of class-switched and non-switched B cells in polymorphic EBV-driven lymphoproliferations appears incompletely defined, at least some have been reported as non-switched [36]. Based on our data, we speculate that the emergence of the EBV-driven B cell populations in the IL6 and APRIL supported PC culture is a particular feature of the progeny of EBV latently infected IgM+ IgD+ NSM B cells. Furthermore, we speculate that the variation in both the absolute number of latently infected memory B cells isolated from each donor and the different memory B cell compartments in which the EBV may reside contributes to the different patterns and frequencies of EBV-associated cells that are observed later in the differentiation. 

A feature of the EBV-associated B cell clusters that we observed was the expression of both *TNFSF13B*-encoded BAFF and *TNFRSF13C,* which encoded one of its cognate receptors, TACI. This provides the potential for an autocrine loop and differential signalling derived from TACI as opposed to BCMA in the context of the APRIL-driven PC survival conditions used in this study. A link between EBV-associated B cell expansion and BAFF expression has been previously described in the context of multiple sclerosis [37]. PC culture conditions also employ IL6 to support PC survival. IL6 has been identified as an autocrine growth factor for EBV-immortalized B cells [38]. Moreover, the targeting of IL6 has shown some efficacy in controlling post-transplant EBV-driven B cell LPDs [39]. Thus, the particular cytokine conditions used to support PC survival may be modified by autocrine loops to support EBV-associated B cell expansion when these emerge following initial B cell activation and differentiation. In the future, it will be interesting to test whether enrichment or depletion of specific subsets can identify B cells most prone to EBV-driven spontaneous transformation, whether this is linked to latently infected IgM+ IgD+ NSM B cells in particular, and whether autocrine loops contribute to their expansion in some donors. 

## 5. Conclusions

In summary, here we presented an analysis using targeted scRNAseq of the emergence of spontaneously arising EBV-driven B cell populations amongst differentiating memory B cells. Our data supports a particular contribution of IgM+ IgD+ NSM B cells to spontaneously arising EBV-driven B cell expansions in vitro, and identifies distinct cell states that vary between donors but follow reproducible patterns linked to B cell activation and differentiation related to those of GC B cells, as also recently reported in immortalized EBV-driven LCLs [24].

## Figures and Tables

**Figure 1 cancers-15-03083-f001:**
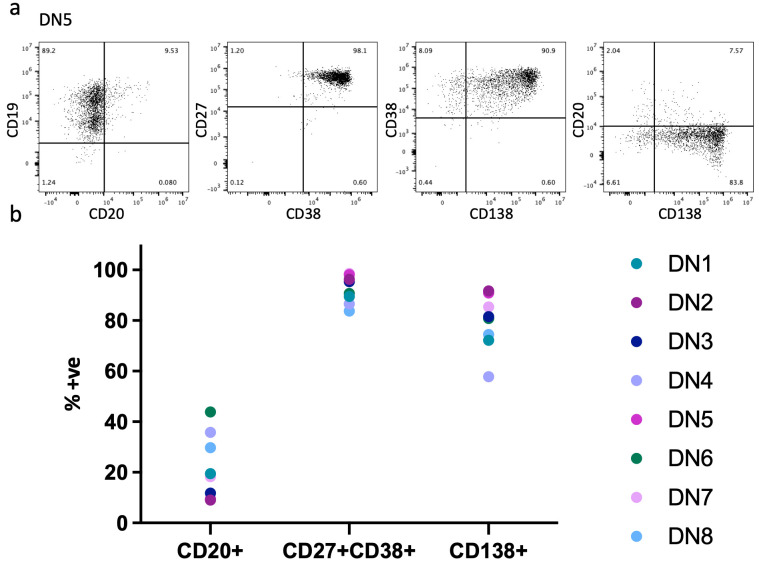
Flow cytometric phenotype of differentiations: (**a**) upper panels show representative flow cytometry data for one donor (DN) at the point of scRNAseq analysis showing CD19 vs. CD20, CD38 vs. CD138, and CD20 vs. CD138, from left to right; (**b**) lower panel summary of phenotypes of all eight donors (labelled DN1-DN8) for CD20+, CD27+, CD38+, and CD138+ quadrant gates, as shown in the upper panel. Detailed flow phenotypes for all eight donors are provided in Appendix A.

**Figure 2 cancers-15-03083-f002:**
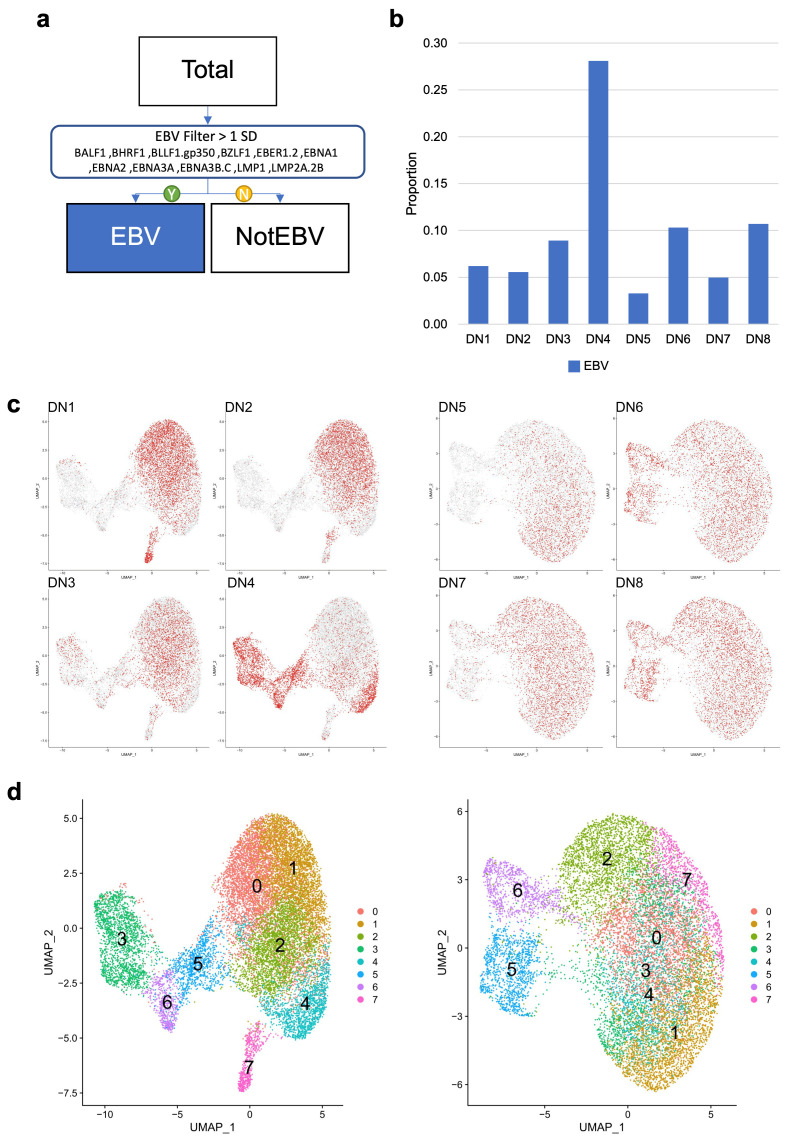
Targeted scRNAseq analysis of all differentiated cells: (**a**) summary of filtering to identify EBV-associated cells in the scRNAseq date; (**b**) bar graph summarizing the fraction of EBV-associated cells identified for each donor (y-axis–percentage proportion; x-axis—donors labelled DN1-DN8); (**c**) UMAP plots for all cells-left panels experiment 1 (E1) and right panels experiment 2 (E2) shown in each of the four panels are the contribution of cells from the indicated individual donors DN1-8 with distribution of donor cells across the UMAP plots shown in red; (**d**) distribution of cell clusters identified in each experiment, illustrated in the colour-coded legend on the right of each panel (E1—left panel; E2—right panel).

**Figure 3 cancers-15-03083-f003:**
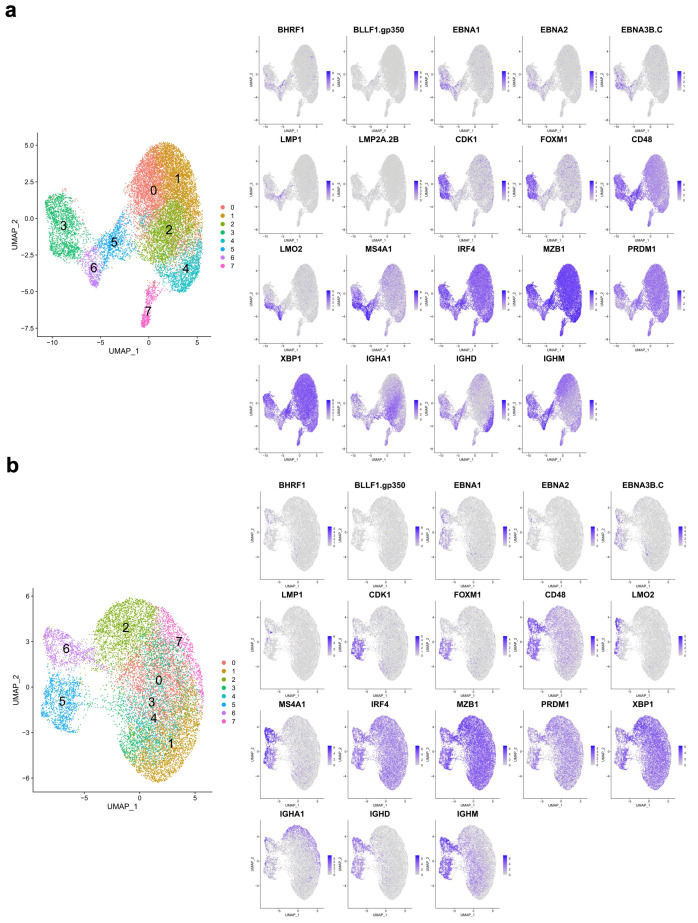
Expression pattern of index genes across all cells: (**a**) expression patterns for index genes as identified in each panel across the UMAP plot for experiment 1; (**b**) equivalent expression pattern across the UMAP plot for experiment 2. Gene identity is indicated with official gene symbol above each panel with expression illustrated on a grey to purple expression scale for each gene.

**Figure 4 cancers-15-03083-f004:**
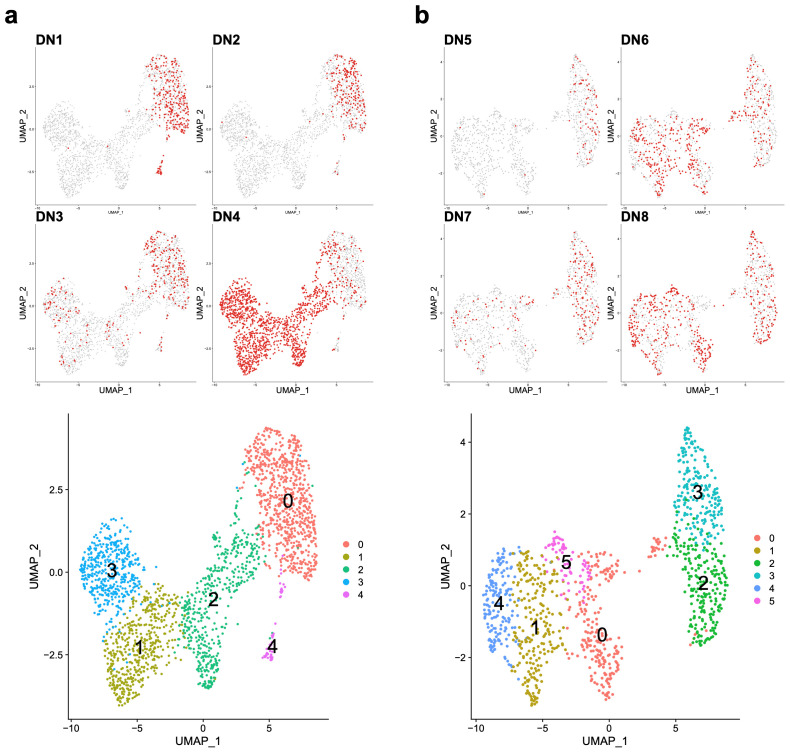
Focused analysis of EBV-associated cells: focused analysis of all EBV-associated cells for experiment 1 (**a**) and experiment 2 (**b**). Upper four panels display the contribution of cells from the indicated individual donors (**a**) DN1-4 and (**b**) DN5-8 with distribution of donor cells across the UMAP plots shown in red. Lower panels show the distribution of identified cell clusters in each experiment identified by the colour-coded legend.

**Figure 5 cancers-15-03083-f005:**
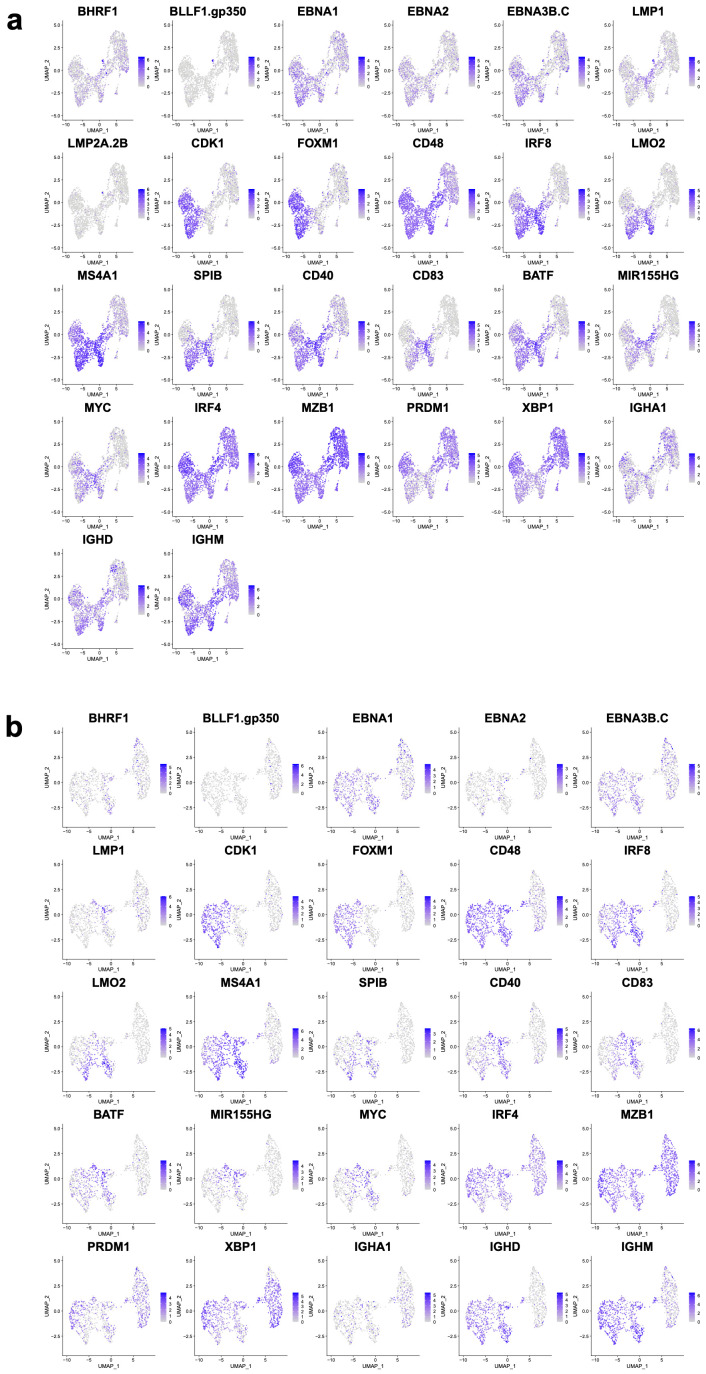
Expression pattern of index genes across EBV-associated cells: (**a**) expression patterns for index genes as identified in each panel across the UMAP plots for EBV-associated cells in experiment 1 and (**b**) equivalent expression pattern across the UMAP plots for experiment 2. Gene identity is indicated with official gene symbol above panel with the expression illustrated on a grey-to-purple expression scale for each gene.

**Figure 6 cancers-15-03083-f006:**
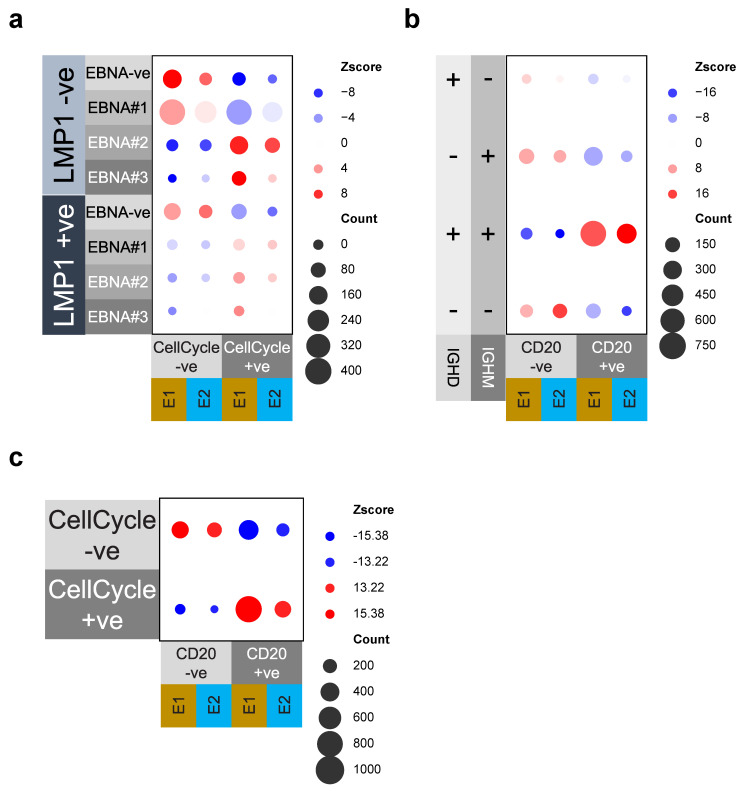
Significant associations between phenotypic and EBV gene features: (**a**) Bubble plot illustrating significance of association between EBV gene expression state (left side of plot) divided between *LMP1*−ve status upper part, and LMP1+ve lower part and then by expression of 0–3 *EBNA* genes, and separated by assigned cell cycle (left cell cycle negative, right cell cycle positive) and separated by experiment. The significance of association for different combinations is illustrated as a Z-score (−8 blue to red +8 colour) and absolute cell count (size 0–400), as illustrated in the legend. (**b**) Bubble plot illustrating the significance of association between the expression of *MS4A1/CD20* and *IGHM,* and/or *IGHD* heavy chains. Illustrated on the left are the combinations of *IGHM* or *IGHD* as the detected expression state; along the bottom is the *MS4A1/CD20* expression status, separated by experiment. The significance of association for different combinations is illustrated as a Z-score (−16 blue to red +16 colour) and absolute cell count (size 0–750), as illustrated in the legend. (**c**) Bubble plot illustrating the significance of the association between cell-cycle status on the left and *MS4A1/CD20* expression along the bottom, divided by experiment. The significance of association for different combinations is illustrated as a Z-score (−15.38/−13.22 blue and red +15.38/13.22) and absolute cell count (size 0–1000), as illustrated in the legend.

**Figure 7 cancers-15-03083-f007:**
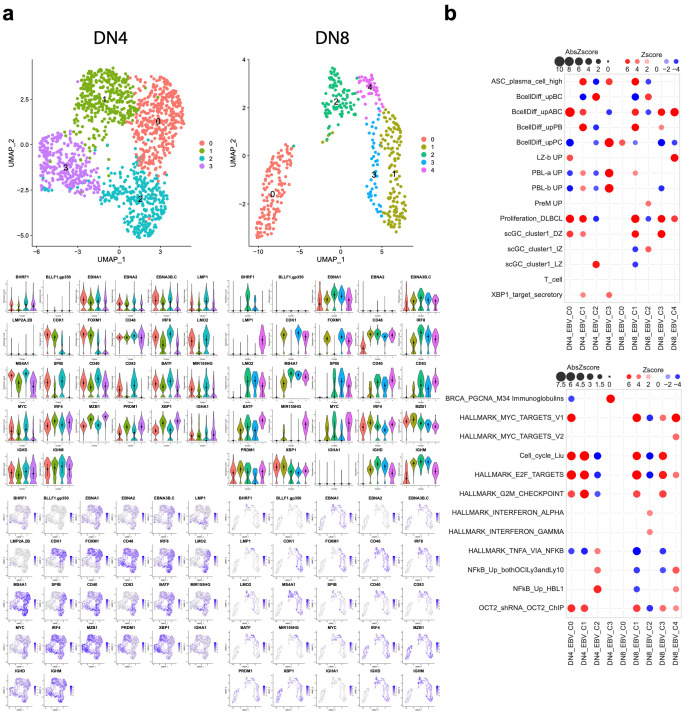
Donor level analysis of EBV-associated cells: (**a**) Analysis of EBV-associated cells for individual donors shown for DN4 (left) and DN8 (right). Each panel is divided to show the UMAP plot with cell clusters identified with colour coding. Beneath this are patterns of index gene expression with each gene shown identified by the official gene symbol and plotted either as violin plots of gene expression divided by cell clusters (middle panels) or expression mapped onto individual cells in the UMAP plots (lower panels). (**b**) The bubble-plot illustrates select signature enrichments across the identified cell clusters for DN4 and DN8. The upper part of figure signature terms relate to B cell differentiation states; the lower part of the figure signature relates to the selected pathways. Enrichment is shown as an absolute Z-score of enrichment (bubble size). The positive-to-negative Z-score is shown on a red-to-blue colour scale (red +6 to blue −4).

## Data Availability

Data sets are available via the Gene Expression Omnibus with accession number GSE232418.

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
