# Peer review of "Spontaneous EBV-Reactivation during B Cell Differentiation as a Model for Polymorphic EBV-Driven Lymphoproliferation"

_cancers, 2023, doi:10.3390/cancers15123083_

Round 1

Reviewer 1 Report

Matthew et al. have worked on the “Spontaneous EBV-reactivation during in vitro B-cell differentiation: as a targeted single cell expression analysis” describing how EBV expression patterns vary in PC during its differentiation. The hypothesis is interesting with the significant application of new techniques like scRNAseq. However, the following major revision need to be improved the overall quality of the manuscript.

1.The title can be modified to “Spontaneous EBV-reactivation during B-cell differentiation as a model for polymorphic EBV-driven lymphoproliferation”

2. An overall descriptive image explaining the findings of the study can make the study much more understandable to the readers.

3.The authors state that they have collected peripheral blood from 8 healthy donors but the authors conclude that “patterns of gene expression in primary EBV reactivation are consistent with recently proposed models relating EBV mediated transformation in lymphoblastoid cell lines”. This statement looks controversial. Because the authors fail to explain the gene patterns in lymphoma patients, experimentally.  How do you explain this? If not, you can rewrite the manuscript supporting your observations.

4.Are the healthy donors clinically tested for their fitness? If they were earlier exposed to any infections (as infections can affect the cell numbers of peripheral blood subtypes)? If so, mention those selection criteria in the methodology.

5.How would you explain cell cycle and MS4A1/CD20 expression have a relationship, at least in the case of healthy donors, without experimental support?

6.The major fault of a manuscript is that author states many things from previous studies or data mining. But they bring healthy cohorts experimentally. This study has a great limitation to conclude the hypothesis that EBV-reactivation causes lymphoproliferation. 

The English should be revised with the native pharmacologist.

Author Response

 1.The title can be modified to “Spontaneous EBV-reactivation during B-cell differentiation as a model for polymorphic EBV-driven lymphoproliferation”

We thank the reviewer for this suggestion and have modified the title accordingly

  1. An overall descriptive image explaining the findings of the study can make the study much more understandable to the readers.

We have provided this in new Supplemental Figure 10

3.The authors state that they have collected peripheral blood from 8 healthy donors but the authors conclude that “patterns of gene expression in primary EBV reactivation are consistent with recently proposed models relating EBV mediated transformation in lymphoblastoid cell lines”. This statement looks controversial. Because the authors fail to explain the gene patterns in lymphoma patients, experimentally.  How do you explain this? If not, you can rewrite the manuscript supporting your observations.

We thank the reviewer for this comment, but we argue that the conclusion we reach is soundly based on the observed data.

In the work cited from the Luftig lab (citation 21) the authors describe the pattern of gene expression at single cell level in EBV LCLs. This pattern of expression led to their suggestions of a GC like state maintained by EBV in LCLs. As we state we have analysed the differentiation of 8 heathy donors none of whom had an intercurrent infection or significant medical condition at the time of blood sampling, and are to our knowledge representative of healthy individuals in our region. When we analyse the cells generated in the in vitro system as we describe in detail, we observe a similar pattern to the gene expression changes described by Luftig et al. Hence we argue that our conclusion is  entirely justified, as we show in various figures signatures of genes related to GC cell states are significantly enriched in different cell populations that we observe and diverge from the predominant plasma cell state.

LCLs have long been used as models of EBV transformation but are derived from healthy donors with laboratory EBV infection and are not derived from “lymphoma patients”. Similarly polymorphic LPDs in patients are not analogous to a lymphoma. If the pathology fits a lymphoma diagnosis then the correct term would be a monomorphic EBV-driven LPD with features of the “lymphoma type” (this is covered in the WHO Haem4R and new Haem5 classifications). By making a comparison to LCL data we are not making a claim that the cells in our model are analogous to a lymphoma state, but instead to EBV immortalised primary B-cells. We do not see this as controversial because the process of primary EBV reactivation and the sustained maintenance of an EBV-driven LCL are intrinsically linked to the biology of EBV as has been widely documented in previous studies. The expression states we observe are very similar to those described by Luftig et al, and that is a conclusion supported by all the data we generate and analyse in the manuscript, we do not see the need at this stage to rerun EBV LCLs on our platform to regenerate data that has already been established elsewhere.

To clarify the difference between an EBV LCL and EBV driven lymphoma cell lines we have modified a sentence in the introduction (lines 74-76).

4.Are the healthy donors clinically tested for their fitness? If they were earlier exposed to any infections (as infections can affect the cell numbers of peripheral blood subtypes)? If so, mention those selection criteria in the methodology.

We agree with the reviewer that this is an interesting point, we have modified the statement regarding the donors on line 103-104 and lines 224-228. We did not test the donors clinically for fitness, as that would be beyond the scope of our study, but these were all otherwise healthy adults approximate age range 25-55 with no known intercurrent infection or medical condition who are representative to the best of our knowledge of healthy adults in our location.

5.How would you explain cell cycle and MS4A1/CD20 expression have a relationship, at least in the case of healthy donors, without experimental support?

We thank the reviewer for this question, we observe that cell cycle is associated with two different cell states as shown in Figure 7b for example at individual donor level, and to further extend the evidence supporting this conclusion we have included new Figure 6c and results on lines 382-389.   Our data overall demonstrate that populations of proliferative cells with MS4A1/CD20 expression are seen in donors with higher EBV cell number in the differentiations.

Why MS4A1/CD20 and cell cycle are associated is a different question for which future studies will be needed. We consider this is likely due to the fact that EBV locks the cells in an activated B-cell like state as a result of signalling pathway activation such as sustained NFkB activation which in other contexts has been identified as capable of blocking differentiation, and at the same time establishing an epigenetic lock on the activated B-cell state as has been seen in LCL cells (refs 18 and 19 in the manuscript). To address this point we have added a short paragraph to the discussion (lines 492-499)

6.The major fault of a manuscript is that author states many things from previous studies or data mining. But they bring healthy cohorts experimentally. This study has a great limitation to conclude the hypothesis that EBV-reactivation causes lymphoproliferation. 

We respectfully disagree with the reviewer that there is a major fault in this. We generate a substantial amount of new data in a model system which has not previously been examined at single cell level.  This identifies patterns of single cell gene expression in EBV associated B-cells spontaneously arising in differentiation.

That healthy adult donors are a suitable source to test frequency of reactivations reasonable given that we know from population level studies that the frequency of latent EBV infection in the healthy adult population in the UK approaches 90% or more (new reference 33). We use a blood sample of 50ml which assuming a 5l blood volume  should encompass 1% of the circulating B-cell pool given the frequency of latent EBV frequency in the memory B-cell pool we would expect to find such cells at low level in the samples.

To our knowledge this is the first such study to address this, but naturally we draw on the extensive body of data from others who have over many years proven that EBV drives B-cell lymphoproliferations.

We think it is beyond doubt that EBV is a primary driver of B-lymphoid proliferations and lymphomas. To argue that an expanding B-cell population expressing EBV genes is not driven by EBV we think is not tenable. Our data align well with previous studies, and we consider this as an important feature supporting the validity of the work.

Comments on the Quality of English Language

The English should be revised.

We have revised the language for clarity and simplicity where possible. We note that RT is a clinical diagnostic consultant haematopathologist, and we have included an additional co-author Roger Owen who is also a consultant haematopathologits/haematologist. All authors are native English speakers. We believe that the manuscript is improved for clarity.

Reviewer 2 Report

The study uses the primary lymphocytes of peripheral blood from eight donors. The study offers a first study on the single cell transcriptomics of B cells in connection with the EBV status.  The authors are experts in B-cell differentiation and this work, using single cell RNAseq studies the issue of EBV reactivation in differentiating lymphocytes. This has been reported in EBV-transformed lymphoblastoid cell lines but not in healthy primary B cells They first identified the small fraction of cells infected by EBV, and within these cells, they identified several clusters of gene expression.

Using the powerful technique of scRNA seq they study the gene expression patterns in the EBV reactivation upon differentiation of B cells form healthy donors. As expected plasma cells gene expression pattern is present in all EBV-infected subpopulations, which (some already knoken) is a small fraction of the total number of B cells. They show that EBV gene expression is associated to specific signatures as that of cell proliferation. The work fails to identify the subset of B-cells most prone to EBV-driven spontaneous transformation, but it is an interesting study and shed new information.

I have some comments:

Lines 239-240: What was the criteria to use 23-27 days of differentiation?  Would the gene expression pattern significantly change with shorter or longer periods of time?. If the snap-shot opf the scRNAseq is taken earlier or later whic would be the expected consequences?

Figure 2: The lettering of the figure is wrong. The data of DN8 is the part b and the right panels (up and low) are the part c

Lines 237: ….”Memory B-cells from two independent sets of 4 healthy adult  donors…” Do this mean that four samples from four donors were pooled?

Figure 6: the UMAP plots correspond to the two pools of mixed donors? Please clarify in the legend.

Line 388: Why such a difference in the number of clusters among donors?

The finding that there among EVB-infected B cells there a subpopulation with and without LMP1 expression is puzzling: Which can be the reason for the segregation in LMP1 expression (and likely other EBV genes)

MINOR

Line 268:  “...(Figure 1d) in both instances primary groups of cells  ...”  Something missing in this sentence

Typo in the legend of figure 3

Author Response

We thank reviewer 2 for their overall assessment of the manuscript.

Specific comments:

Lines 239-240: What was the criteria to use 23-27 days of differentiation?  Would the gene expression pattern significantly change with shorter or longer periods of time?. If the snap-shot of the scRNAseq is taken earlier or later whic would be the expected consequences?

We have included a statement on line 239-241 to address the choice of time point. Essentially at this time frame our previous bulk population data indicates that the average cell state in the PC differentiation is for cells to have adopted a mature PC state in terms of secretory gene expression and to have exited cell cycle. Therefore, if EBV was driving a proliferating and or B-cell phenotype cell state it would be readily identified against the background PCs. At earlier timepoints prior to day 13 there would be more expected contribution from non-EBV cells to cell cycle and B-cell phenotypes.

Regarding changes at other time points this is a very interesting question. We agree absolutely that the snap-shot imposes limitations. It does not provide the information that a time course would and cannot tell us whether the low EBV cell number donors would eventually generate expanded B-cell populations. That said we have taken many differentiations well beyond day 30, while we have not analysed these with single cell expression we can anecdotally report that while a few of these generate cell clusters in keeping with transformation by EBV the majority do not. Thus, we think that the day 23-27 state is indicative of the future fate of the culture. Because these data are anecdotal and not systematic we have not included them in the manuscript, but we are keen to develop this aspect in future work.

We have added a paragraph on lines 502-510 to acknowledge this interesting point.

Figure 2: The lettering of the figure is wrong. The data of DN8 is the part b and the right panels (up and low) are the part C

We apologies for some confusion in the original figure legends and numbering we have amended.

Lines 237: ….”Memory B-cells from two independent sets of 4 healthy adult  donors…” Do this mean that four samples from four donors were pooled?

We agree this wording was confusing and have amended. It is 8 donors divided between two experiments each with 4 donors. Lines 232-233 have been amended to clarify this.

Figure 6: the UMAP plots correspond to the two pools of mixed donors? Please clarify in the legend.

We apologise that the formatting of Figures with legends the order was confused and was not noted prior to submission. This has now been amended Figure 2 and 3 are the pooled experiments showing all cells, Figure 4 and 5 are the pooled data for EBV+ cells only.

Line 388: Why such a difference in the number of clusters among donors?

The number of observed clusters is linked to the absolute number of EBV-associated cells. As these increase the heterogeneity increases generating 2-5 cell clusters. In donors with high cell number 4 main cell states become apparent as shown for DN4 and DN8. We reason that the differences relate to the expected variation between the number of EBV latently infected cells in different donors and the variation in type of memory B-cell in which the latency is maintained which is reflected in the B-cells seeded into the culture at day 0.

We have added a brief sentence to address this in the discussion on line 559-562.

The finding that there among EVB-infected B cells there a subpopulation with and without LMP1 expression is puzzling: Which can be the reason for the segregation in LMP1 expression (and likely other EBV genes)

While the number of LMP1 expressing B-cells appears relatively small we think this reasonably reflects the situation also observed in LCLs and in EBV-driven pathologies. This is consistent with a mixed distribution of EBV latency states in the population.  

The Luftig lab have demonstrated a similar variation in LMP1 expression in LCLs analysed with single cell approaches. In the Luftig data (https://doi.org/10.3389/fimmu.2022.1001145 Figure 3E in that manuscript) which is derived from LCLs on a different scRNAseq platform it is also apparent that LMP1 expression is relatively rare. The idea that EBV-driven cell expansions is maintained by cells in just one EBV latency state is somewhat challenged by their and our data, but that may reflect the fundamental difference between analysing bulk populations and single cells. In terms of pathology there are definite parallels where frequent discrepancies are seen between the extent of EBV detection with EBER-ISH (very sensitive all EBV latency states) and EBV LMP1 IHC.

The model proposed by Luftig and colleagues that our data support is that the populations transition between the LMP1 expressing cell state that promotes growth and limits division (GC LZ like) and the LMP1 negative state which is the latency IIb like state in which the cells proliferate very profoundly (GC DZ like).

MINOR

Line 268:  “...(Figure 1d) in both instances primary groups of cells  ...”  Something missing in this sentence

We have reviewed and confirm the line now reads.

“Both sets of samples resolved into 8 clusters (Figure 2d) in both instances two primary groups of cells were resolved the larger encompassing clusters C0-2+4 (experiment 1) or C0-4+7 (experiment 2).”

Typo in legend 3

We have edited the legend.

Reviewer 3 Report

In the manuscript entitled "Epstein-Barr Virus-Associated Cancers: From Pathogenesis to Treatment"  the authors employed single cell RNA seq technique to determine the differential expression of genes during PC differentiation. the authors also identified a population of EBV gene expressing cells and B cell expression features during differentiation. The manuscript is very nicely drafted and presented in a lucid way that makes it easy to understand. The manuscript touched all the aspects and to my understanding the manuscript is fit for publication in the the current for with some minor english editing (if required)

The overall English quality is good but may need some special editing if required.

Author Response

We thank the reviewer for their comments. The authors have reviewed the language and layout of the manuscript and adjusted text for clarity in lines with other comments received.

Reviewer 4 Report

This research performed targeted single-cell RNAseq analysis using the BD Rapsody analysis system on lymphocytes of 8 donors. I understood (and I may be wrong) that the analysis look for genes associated to the different latency and stages of differentiation in the context of EBV infection. The research is very interesting, and relevant, and uses state-of-the-art methodology. Nevertheless, the presentation of the data could be improved and/or summarized using tables and/or figures. I would recommend making a figure regarding the methodology, with a diagram of the experimental design. And a final Table with the summary of the most relevant findings. After reading the manuscript, it was not clear to me if all donors were EBV-positive.

Additional comments:

1) In the introduction. Could you please add a figure showing the different stages of B-cell differentiation and the expression of the different latency markers of EBV?

2) In the introduction, is it possible to describe the different diseases associated with EBV?

3) Does EBV only infect B-cells?

4) Section 2.1. Could you please add the catalogue number of the different antibodies?

5) Lin 98. Do you know the characteristics of the 8 healthy donors? By "healthy" do you mean EBV negative?

6) Line 102. Why CD23 is used to enrich memory B-cells?

7) Section 2.3. Could you please remind the readers why APRIL was used?

8) Could you please upload the Fastq files to a repository?

9) Is "SD" (line 152) the standard deviation?

10) In Figure 1B. Is "DN" a donor? Then,  are all of them EBV positive?

11) In the UMAP images, are the different cluster of cells from 0 to 7 according to the colors?

12) Lines 173-189. Is the 43,572 final signature available?

13) Regarding 2.9. Was the C8: cell type signature gene sets included as well?

14) Line 192. Could you please add "CD20" next to MS4A1? Could you please also add "constant region of heavy chain of IgD or IgM"?

15) Line 193. Could you please add "cell division cycle 2, G1 to S and G2 to M" next to CDK1? In a similar manner, " M-phase phosphoprotein 2", and "marker of proliferation Ki-67" for FOXM1 and MKI67, respectively.

16) Section 2.10. Did you also analyzed EBERs and miRNAs? How did you identified healthy EBV carrier cells (latency 0)?

17) Section 2.12. Is "07/Q1206/47" the IRB code?

18) This research uses UMAP plots. Could you please explain (if feasible), why UMAP was used instead of t-SNE plots? Is it because of the equipment software?

19) How many genes are analyzed by the BD Rhapsody analysis system? Is it 701 genes as shown in 223?

Author Response

We thank the reviewer for their comments. We have addressed these as follows

 Comments:

1) In the introduction. Could you please add a figure showing the different stages of B-cell differentiation and the expression of the different latency markers of EBV?

We thank the reviewer for the suggestion. We have included a new summary figure (Supplemental Figure 10) in which we include a summary assessment of the latency markers associated with the different B-cell/PC states.

2) In the introduction, is it possible to describe the different diseases associated with EBV?

While we have not included an exhaustive listing, we have mentioned specific lymphoma subtypes and some other cancers linked to EBV on new lines 37-41

3) Does EBV only infect B-cells?

The B-cell lineage is the primary lineage for latent EBV maintenance. Epithelial infection is a presumed natural component of viral entry and linked to rare epithelial malignancies, similarly the association of EBV with NK/T-cell lymphoma nasal type implies rare infection of this lineage.

4) Section 2.1. Could you please add the catalogue number of the different antibodies?

These have been included in lines 129-135.

5) Lin 98. Do you know the characteristics of the 8 healthy donors? By "healthy" do you mean EBV negative?

We have addressed this point more clearly in text and methods Lines 103-104 and 229-230.

By healthy we mean adult donors drawing from the volunteering adult population in our location, approximate age range 25-55, who at the time of blood donation were not suffering from an intercurrent infection and did not have a known relevant health condition.

To clarify previous studies estimate that 90% of the adult population is latently infected with EBV (new reference 33 for recent data on UK population). Hence the majority of all “healthy” individuals however defined are likely to have EBV latency present. This expected high frequency of latent infection is consistent with the fact that all 8 donors showed some evidence of EBV presence on memory B-cell differentiation.

6) Line 102. Why CD23 is used to enrich memory B-cells?

We have clarified this further in the methods (line 106-109). This is a standard negative selection approach. CD23 is used to deplete naïve B-cells from the B-cell pool. The reason to take this approach rather than a positive selection is that sequential negative selection leaves the memory B-cell population behind as an “untouched” population with less impact of the purificiation on the memory B-cells. For example if CD27 is used to positively select memory B-cells, this ligates CD27 prior to in vitro activation and the CD27 antibody used in positive selection remains attached to the differentiating B-cells across several subsequent cell divisions.

7) Section 2.3. Could you please remind the readers why APRIL was used?

We have added this on line 235-236.

8) Could you please upload the Fastq files to a repository?

This has been actioned data is available with accession number GSE232418 (line 149-150 and in new data availability statement).

9) Is "SD" (line 152) the standard deviation?

Yes that is correct we have now expanded this.

10) In Figure 1B. Is "DN" a donor? Then,  are all of them EBV positive?

Yes DN is the abbreviation we used for donor. We apologise this was not sufficiently clear and have added this to the text and legend.

It is correct that the final analysis finds that all the donors had evidence of EBV when analysed in the differentiation. We did not test the donors for presence of EBV prior to differentiation.

11) In the UMAP images, are the different cluster of cells from 0 to 7 according to the colors?

Yes this is correct, apologies this was not clearly stated in the legend, we have amended this.

12) Lines 173-189. Is the 43,572 final signature available?

While 43,572 terms were generated by merging the gene lists, only those signatures with some degree of overlap with the targeted 701 gene panel can give meaningful results. Those signatures with no gene overlap with the targeted panel are not included. We only provide listings for signatures with overlap with the targeted panel in the supplement (a sentence clarifying this has been added in the methods line 188-189).

13) Regarding 2.9. Was the C8: cell type signature gene sets included as well?

Yes the C8 cell type signatures from the Broad are included and are available in the supplemental data tables

14) Line 192. Could you please add "CD20" next to MS4A1? Could you please also add "constant region of heavy chain of IgD or IgM"?

We have amended new line 195

15) Line 193. Could you please add "cell division cycle 2, G1 to S and G2 to M" next to CDK1? In a similar manner, " M-phase phosphoprotein 2", and "marker of proliferation Ki-67" for FOXM1 and MKI67, respectively.

We have amended new lines 196-198

16) Section 2.10. Did you also analyzed EBERs and miRNAs? How did you identified healthy EBV carrier cells (latency 0)?

We included EBER but not miRNAs in panel design, in part because the BD Rhapsody method. However EBER1/2 was not adequately detected to allow conclusions. We can therefore not identify latency 0 cell states in the differentiation. We have now included a brief comment relating to this point line 305-306.

We did not screen for EBV latency 0 in donors or proactively identify carriers. We simply differentiated memory B-cells from 8 healthy donors. We expected some donors to generate EBV+ cells. We did not necessarily expect all 8 to do so. But there is a high prevalence of latent EBV infection in the adult population in the UK (EBV seropositivity predicted to be 85%+ by age 15 in the UK e.g.  BMC Public Health 20, 912 (2020). https://doi.org/10.1186/s12889-020-09049-x). We use a 50ml blood sample volume. Estimating an approximate 5l adult circulating blood volume, we calculate that we sample 1% of the circulating B-cell population. Previous estimates in total B-cells indicated frequencies of between 5 – 380 per 107 B-cells. We differentiate on average 5x106 memory B-cells. Even at the low frequency range EBV latently infected memory B-cells would be expected by chance to be present in the samples. We will need future studies to determine the extent to which the basal EBV B-cell frequency relates to the subsequent phenotypes observed in differentiation.

17) Section 2.12. Is "07/Q1206/47" the IRB code?

Yes this is the original “IRB code” for our ethical approval in the UK this is the research ethics committee (REC) reference number.

18) This research uses UMAP plots. Could you please explain (if feasible), why UMAP was used instead of t-SNE plots? Is it because of the equipment software?

It is our understanding that t-SNE and UMAP provided different ways of addressing similar features in single cell data. We do not seek to make a judgment but have deployed UMAP in our analyses based for example on the work of Becht et al (doi:10.1038/nbt.4314)

19) How many genes are analyzed by the BD Rhapsody analysis system? Is it 701 genes as shown in 223?

Yes this is correct in this instance, but the BD Rhapsody system allows different custom panel design. 701 is toward the upper end of tested custom panels to our knowledge.

Reviewer 5 Report

The authors have performed an intriguing and well-executed study of the diversity found in memory B cells after in vitro expansion, with particular attention to the effects of EBV. I have no suggestions for improvement.

Author Response

We thank the reviewer for their assessment of the manuscript.

Round 2

Reviewer 1 Report

The present revision (Spontaneous EBV-reactivation during in vitro B-cell differentiation: a model for polymorphic EBV-driven lymphoproliferations studied with targeted  single cell expression analys) should be accepted.

No.